# Conversion of Carbon Monoxide into Methanol on Alumina-Supported Cobalt Catalyst: Role of the Support and Reaction Mechanism—A Theoretical Study

**Nguyen Ngoc Ha, Nguyen Thi Thu Ha, Nguyen Binh Long and Le Minh Cam \***

Faculty of Chemistry, Hanoi National University of Education, Ha Noi 100000, Vietnam;
hann@hnue.edu.vn (N.N.H.); ntt.ha@hnue.edu.vn (N.T.T.H.); binhlongsonla@gmail.com (N.B.L.)

\* Correspondence: camlm@hnue.edu.vn; Tel.: +84-989-886-242

**Abstract:** Density functional theory (DFT) was used to calculate the step-by-step hydrogenation of carbon monoxide (CO) to form methanol over a $Co_4$ cluster/$Al_2O_3$ surface. A three-dimensional $Co_4$ tetrahedral structure was selected to explore its interaction with the supporting $Al_2O_3$ (104) surface. $Co_4$ chemically reacted with $Al_2O_3$ to form a new chemical system. The calculated results show that $Al_2O_3$ support has strengthened the $Co_4$ catalyst during the reaction since the formation of the Co–O bond. Loading $Co_4$ on the $Al_2O_3$ surface increases CO adsorption ability but decreases the dissociation ability of C–O to produce hydrocarbons. As such, $CH_3OH$ formation becomes more favorable both kinetically and thermodynamically on $Co_4/Al_2O_3$. In CO hydrogenation, methanol was synthesized through a CO reaction with hydrogen via either an Eley–Rideal or Langmuir–Hinshelwood pathway to form the intermediates C\*-O-H, H-C\*-OH, $H_2$-C\*-OH, and finally the hydrogenation of $H_2$-C\*-OH to methanol with both hydrogenation steps forming C\*-OH and final product as rate-limiting. These results showed that the interaction between Co, $Al_2O_3$ and $H_2$ pressure can change the pathway of CO hydrogenation on Co/$Al_2O_3$ and it may, therefore, influence distribution of the final products.

**Keywords:** density functional theory; $Co_4$ cluster; $Al_2O_3$; support; adsorption; mechanism; methanol

## 1. Introduction

Methanol synthesis by carbon monoxide (CO) and carbon dioxide ($CO_2$) hydrogenation has been widely studied both experimentally and theoretically [1–6] because methanol is an important raw material of crucial importance in the chemical and energy industries.

In recent years oxide-supported metal catalysts have been extensively studied for CO hydrogenation to methanol such as modified catalysts of the Fischer–Tropsch process or Cu-based catalysts [7–12], Mo-based catalysts [13–16], and cobalt nanoparticles [17–20]. However, the methanol selectivity and catalytic stability remain rather poor. Thus, the development of a good catalyst with high selectivity and stability is required. In order to improve the methanol yield, many efforts have been made to understand the reaction mechanism and to suggest different reaction pathways to gain methanol via hydrogenation of CO and $CO_2$.

Many studies have been devoted to the reaction mechanism over catalysts, controversies still remain. In the work done by Li et al. [21] a reaction mechanism on the transition metal (Fe, Co or Ni) promoted $MoS_2$- based catalyst was proposed. Different reaction pathways on different catalytic phases ($MS_x$ and M-KMoS) were suggested. On the $MS_x$ phase, CO dissociates and is subsequently hydrogenated into $CH_x$ and methane while on the mixed M-KMoS phase non-dissociative adsorbed

CO inserts into a metal–methyl carbon bond producing an oxygenate precursor, which upon further hydrogenation or dehydration forms mixed $C^{2+}$ oxygenates and hydrocarbons.

CO hydrogenation on these two $MoS_2$ (10-10) surfaces was also investigated by Shi et al. [22] using density functional theory (DFT) calculations. Their results show that the route for the CO hydrogenation is the following:

$$CO + H \rightarrow HCO$$

$$HCO + H \rightarrow H_2CO$$

$$H_2CO + H \rightarrow H_2COH$$

$$H_2COH \rightarrow OH + CH_2$$

$$CH_2 + H \rightarrow CH_3$$

$$CH_3 + H \rightarrow CH_4$$

Studt et al. [11] proposed that the hydrogenation of CO occurs at the carbon end of adsorbed CO. Firstly, HCO is formed following by $H_2CO$ and finally to $H_3CO$. The last step in the reaction cascade to yield methanol is the hydrogenation of methoxy. Meanwhile, the hydrogenation of the oxygen end generating COH, HCOH and $H_2COH$ intermediates possessed much higher barriers and all intermediates have higher energy than their carbon hydrogenated counterparts

Besides controversy about the mechanism, there is also no general agreement in the nature of the active site(s) and the effect of oxide support. It has been well established that due to the strong metal–support interactions a support can remarkable affect the active metal dispersion, reducibility, which then have an important influence on the catalytic activity [23]. Ramírez et al. [23] who studied the behavior of cobalt-based catalysts supported on several metal oxides in the $CO_2$ methanation reaction at atmospheric pressure and low temperatures (200–300 °C) found that apart from the Co particle size and oxidation state of active species, the catalytic activity and selectivity can also be affected by the nature of the support. Zuo et al. [24] suggested while hydrogenation of CO over metallic Cu does not generate methanol, the methanol can be synthesized from CO hydrogenation over Cu/ZnO due to a strong synergetic interaction between metallic Cu and the oxide support ZnO. In the work done by Lee et al. [25] the effect of the support and the interaction of Cu species with different supports were studied. The authors indicated that over zinc supported catalysts a higher conversion (70%) and selectivity towards higher alcohols (70%) were obtained. However, Medford et al. [26] indicated that the reaction was inhibited by the presence of very low $CO_2$ concentration.

$\gamma$-$Al_2O_3$ is the most common inorganic oxide which is used as a support since its excellent thermal stability, fine particle size, high surface area and wide range of chemical, physical, and catalytic properties. In the work of Prasad et al. [27], iron-based catalysts dispersed on different supports have been extensively investigated. Among studied supports, the performance of $\gamma$-$Al_2O_3$ is best, followed by silicas and titanias. The good performance of may be attributed to the strong metal-support interaction, which results in well dispersed catalyst and hinders catalyst sintering. Reuel et al. [28] studied 10 wt.% Co catalysts supported on MgO, carbon, $SiO_2$ and $Al_2O_3$. They found that the specific activity in the hydrogenation of carbon monoxide depends on the nature of the support in the following order: MgO < carbon < $SiO_2$< $Al_2O_3$.

Theoretically, the mechanism and reaction pathways of CO hydrogenation are also extensively studied. Recently, theory has been recognized as an important tool in the design and optimization of catalysts for many applications [29,30]. Among theoretical methods used to study the electronic configurations and properties of many-body systems, in particular catalytic reactions, DFT is known as a versatile and effective method. The fundamental aspect of the DFT method is that the characteristics of many-electron systems can be using the electron density which depends only on the three Cartesian variables. Using the DFT approach, computational accuracy can be significantly increased without the additional increase in the computational costs. Due to its advances, DFT can simulate catalytic

process at surfaces with the detail and accuracy required for calculated results to compare with experiments [31]. This understanding allows theoretical optimization for better catalysts.

Using periodic DFT calculations Andersen et al. [15] investigated the influence of K-doping on $MoS_2$ activity in the CO hydrogenation. They found that doping K created K-O and K-C bonding on the $MoS_2$ surface and hence enhanced the CO adsorption. The DFT calculations show that K-doping promotes the formation of mixed higher $C_{2+}$ oxygenates. Reimers et al. [30] studied theoretically the catalytic activities of Zn, Ce and Ga oxides in the consecutive hydrogenation reactions of the CO molecule. Their calculated results indicated that the methanol formation proceeds via formyl (HCO), followed by the formation of formaldehyde ($H_2CO$), then methoxy ($H_3CO$) and, finally, methanol.

In the study of Studt et al. [11] DFT results revealed that over Cu (211) the CO hydrogenation occurs at the carbon end of adsorbed CO and the formation of methanol is processed through HCO, $H_2CO$, $H_3CO$ intermediates. In contrast, hydrogenation at the oxygen end will process via COH, HCOH and $H_2COH$ intermediates with much higher barriers.

A DFT study has been conducted by Dou et al. [32] to study the effect of $ZrO_2$ support on the catalytic behavior of $In_2O_3$ in the methanol synthesis from $CO_2$ and CO hydrogenation. The calculations show that $ZrO_2$ has a significant influence on the suppression of the dissociation of $CO_2$ and stabilization of $H_2COO$ species on the surface of $In_2O_3$ catalyst.

The hydrogenation and dissociation reaction network of carbon monoxide (CO) over a Co-doped Cu (111) surface were theoretically studied using DFT. The calculated results revealed that the H-assisted route proceeds with the formation of HCO, $CH_2O$ and $CH_3O$ while OH-assisted route proceeds with the formation of COH, CHOH and $CH_2OH$ [33].

The influence of surface hydroxyls over $Ni/\gamma$-$Al_2O_3$ and $Cu/\gamma$-$Al_2O_3$ has been investigated theoretically by Pan et al. [34] and Zhang et al. [35]. Their calculations show that the hydroxylation of $\gamma$-$Al_2O_3$ changes the reaction pathway and finally will influence the products distribution and hence, the process selectivity.

Zuo et al. [36] investigated theoretically the CO and $CO_2$ hydrogenation on $Cu/\gamma$-$Al_2O_3$ (110) surface in liquid paraffin. Their results indicated that the synthesis of methanol occurs via the formation of CHO, $CH_2O$, and $CH_3O$ intermediates with CHO hydrogenation as the rate-limiting step.

With the above analysis, the objective in this work is to study the impact of the nature of the support on the physicochemical properties of Co-based catalyst and its activity in CO hydrogenation. We investigated the elementary steps resulting in the formation of methanol on $Co/Al_2O_3$ using the density functional theory (DFT) calculations. $Co_4$ cluster was chosen because $Co_4$ is the smallest cluster that possesses a three-dimensional structure and it could include both metal–metal and metal–support interactions. A model of $Al_2O_3$(104)-(2x2x2) based on X-ray diffraction (XRD) experimental results was adopted [37].

In our results, firstly we concentrate on the reactants adsorption and the configurations of the key intermediates involved in the process mechanism, followed by the energies.

## 2. Model and Computational Methods

In this study, a model of $Al_2O_3$(104)-(2x2x2) was selected based on the XRD experimental results [37]. Clusters of $Co_4$ in tetrahedral and rhombus forms were studied to select the stable structure.

The SIESTA (Spanish Initiative for Electronic Simulations with Thousands of Atoms) package based on the DFT approach was used for all geometry and energy calculations [38]. SIESTA has been successfully performed to study on the Fischer–Tropsch reaction [39–41] due to its advantages in robust and accurate aspect.

The generalized gradient approximation (GGA) with the Perdew, Burke, and Ernzerhof (PBE) non-local gradient-corrected functional was employed to estimate the exchange correlation energy [42]. According to the large number of atoms in the studied systems (about 100 atoms), the double zeta basis plus polarization orbitals (DZP) was used for valence electrons, while the core electrons were treated using the norm-conserving pseudo potentials (NCP) in its fully non-local (Kleinman–Bylander)

form [43]. The Coulomb potential was expanded in a plane-wave basis with an energy cut-off of 150 Ryd. The systems were placed in boxes of $25 \times 25 \times 25$ Å, which are big enough to have negligible electric fields at their edges. Spin-polarized calculations have been performed for all systems including metals. All equilibrium structures were obtained using the quasi-Newton algorithm, and the forces acting on the dynamic atoms all are smaller than 0.05 eV/Å.

When studying the step-by-step conversion of CO into methanol on the catalytic system, all the transition states were determined using a climbing image nudged elastic band (CI-NEB) method [44]. The advantages of the CI-NEB methods are that after the convergence is reached, the climbing image will converge to the saddle point and all the images occuring in the reaction coordinates are being relaxed simultaneously. In this work, the total number of images involved in the reaction path is seven, including the initial and the final configurations. In the CI-NEB calculations, the convergence tolerance is 0.1 eV/Å in the magnitude of the forces.

The energy variation is used as a significant criterion to predict the ability and the extent of all the processes. Additionally, to estimate the nature of the adsorbate and substrate interaction, a crucial change in the geometrical parameters was analyzed. Moreover, the atomic partial charges, estimated by means of the Voronoi deformation density (VDD) method, were reported. The VDD method avoids the problems inherent to basis set based schemes and provides meaningful charges that conform to chemical experience [45]. Moreover, the Mayer bond orders were also calculated. The Mayer bond orders are also less dependent on the basis of set choice and they are transferable and can be used to describe similar systems.

## 3. Results and Discussion

### 3.1. Electronic Properties of $Co_4$ and $Co_4/Al_2O_3$ Systems

$Co_4$ clusters were optimized in both tetrahedral and rhombus geometries (Figure 1). The calculated binding energies, $E_b$ ($E_b = [4E(M) - E(M_4)]/4$) and the numbers of unpaired electrons, $N_{ue}$ are presented in Table 1.

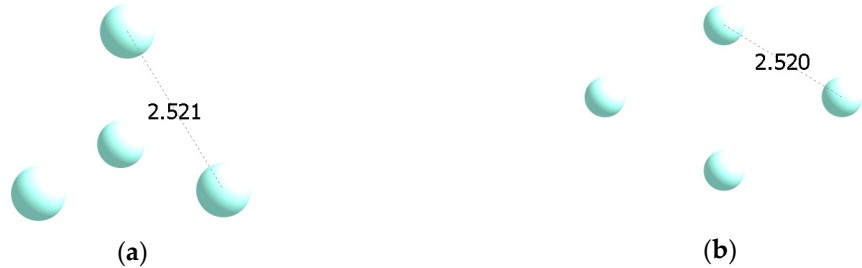

| (a) | (b) |

**Figure 1.** Tetrahedral (**a**) and rhombus (**b**) structures of $Co_4$ cluster; key distances are given in Å.

**Table 1.** The calculated binding energies ($E_b$, eV/atom), the numbers of unpaired electrons ($N_{ue}$) of $Co_4$ clusters.

| Structure | $E_b$ | $N_{ue}$ |
|---|---|---|
| Tetrahedral | 3.86 | 12 |
| Rhombus | 3.41 | 12 |

The calculated binding energy of the tetrahedral structure is higher than that of the rhombus. Hence, for $Co_4$ cluster the tetrahedral structure is more pronounced.

In our calculations, we have used collinear spin polarized option. The total spin polarization for the optimal structure were automatically determined corresponding to the structure with the lowest energy. Furthermore, the $Co_4$ structure were optimized with the fixed spin corresponding to the presence of 10 or 14 unpaired electrons. The results have confirmed that the $Co_4$ structure with 12 unpaired electrons is the most stable due to the lowest energy (see Supplementary, Table S1).

It is interesting to note that the Co atom has three unpaired electrons, while $Co_4$ has 12 unpaired electrons. Thus, the formation of $Co_4$ does not involve the electron pairing. In other words, metallic bonding is present even in clusters as small as $Co_4$. Moreover, the Co–Co bond lengths in tetrahedral and rhombus clusters are 2.521 and 2.520 Å, respectively which are shorter than twice the atomic radii of Co (1.35 Å, [46]). This again confirms the formation of chemical bonds between cobalt atoms in the cluster structures.

To evaluate the interaction between the clusters and the support, the initial cobalt clusters were placed at several positions on the surface of $Al_2O_3$ and the interaction energies were calculated (See Supplementary, Table S2). By comparing the interaction energy, the most stable structure of $Co_4$ on $Al_2O_3$ was found and is presented in Figure 2.

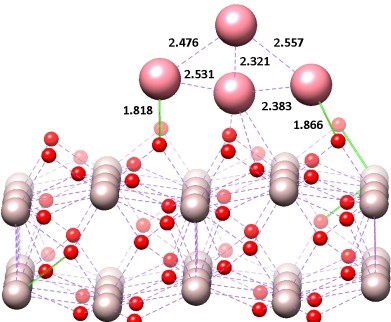

**Figure 2.** The optimized structure of $Co_4/Al_2O_3$ systems (colors: red—oxygen atom, grey—alumina atoms, violet—cobalt atoms); key distances are given in Å.

The structure of the $Co_4$ cluster on $Al_2O_3$ is close to the tetrahedron, the number of unpaired electrons of $Co_4$ has been reduced to 10 in comparison to the cluster alone. This is attributed to the partly transfer of electrons from the metal clusters to the alumina support. The total atomic charge of $Co_4$ cluster is calculated to be +0.944 which demonstrates the strong interaction between the cluster and the support. Moreover, the closest distance between Co and O atoms in $Co_4/Al_2O_3$ system (1.818 Å) is shorter than the sum of the atomic radius of Co and O atoms (1.95 Å [46]). Hence, the formation of $Co_4/Al_2O_3$ system is certainly accompanied by the formation of chemical bonds between Co and O atoms in $Al_2O_3$. The bonding formation is confirmed by the total Mayer bond order between Co and O, which is determined to be 1.712.

Due to the formation of chemical bonds the properties of $Co_4/Al_2O_3$ system are expected to differ from initial $Co_4$ cluster.

### 3.2. Adsorption of CO and $H_2$ on $Co_4$, $Al_2O_3$ and $Co_4/Al_2O_3$ Systems

#### 3.2.1. Adsorption of CO on $Co_4$ and $Co_4/Al_2O_3$ Systems

The adsorption of CO on the catalytic systems plays an important role and mainly governs the formation of hydrogenation products. Therefore, a detailed study of the mechanism of this process is essential.

Adsorption of the CO molecule on transition metals can be defined in terms of donation of electron density from CO to the metal and $\pi$-back donation of electrons from filled *d* orbitals on the metal into vacant antibonding $\pi^*$ orbitals on the CO [47]. Hence, CO is more favorably adsorbed via the C atom than the O atom. The calculated results also indicate that when a CO molecule is adsorbed on $Co_4$ and $Co_4/Al_2O_3$ systems the interaction of C–Co is always thermodynamically more favorable compared to the O–Co orientation (see Supplementary, Table S3). With the C–Co orientation, there are two possibilities for the CO adsorption: C atom in CO is bound to one Co atom (d-1 configuration); and C atom in CO is simultaneously bound to two Co atoms (d-2 configuration). The optimized configurations of CO adsorbed on $Co_4$ cluster and $Co_4/Al_2O_3$ system are presented in Table S3 of Supplementary.

The optimized structures of the d-1 and d-2 configurations for the CO adsorbed on $Co_4/Al_2O_3$ are illustrated in Figure 3. Table 2 summarizes the calculated results of CO adsorption on $Co_4$ and $Co_4/Al_2O_3$ systems.

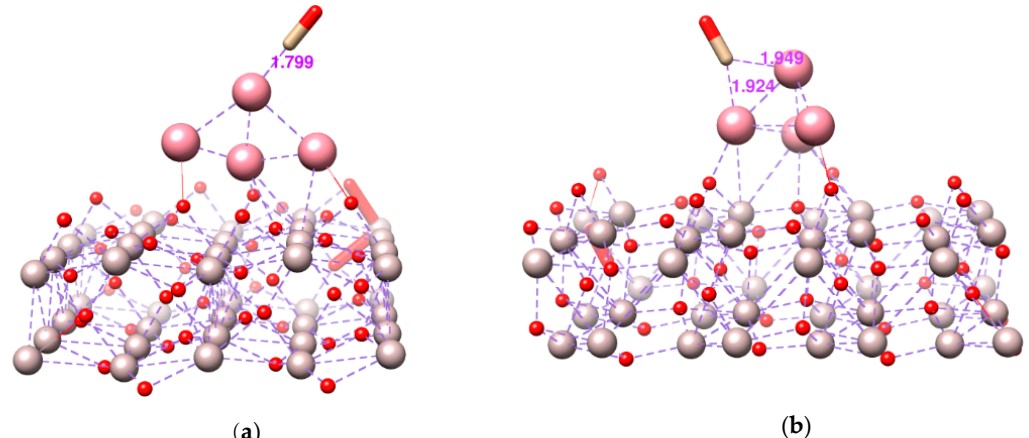

(**a**)                                                                              (**b**)

**Figure 3.** The optimized structure of the d-1 (**a**) and d-2 (**b**) configurations of CO-$Co_4/Al_2O_3$; (colors: red—oxygen atom, grey—alumina atoms, violet—cobalt atoms); key distances are given in Å.

**Table 2.** $E_{ads}$ (in kJ mol$^{-1}$), $d_{C-O}$ (in Å) and bond-dissociation energy (BDE) of C–O (in kJ mol$^{-1}$).

| Structures | d-1 Configuration | | | d-2 Configuration | | |
|---|---|---|---|---|---|---|
| | $E_{ads}$ | $d_{C-O}$ | $BDE_{C-O}$ | $E_{ads}$ | $d_{C-O}$ | $BDE_{C-O}$ |
| $Co_4$ | −186.9 | 1.172 | 960.6 | −201.4 | 1.203 | 928.0 |
| $Co_4/Al_2O_3$ | −231.9 | 1.161 | 1004.3 | −237.3 | 1.179 | 1023.5 |
| CO | | 1.145 | 1190.3 | | 1.145 | 1190.3 |
| Exp. | | 1.128 [48] | 1072 [49] | | | |

From Table 2, one can see that when CO molecule is adsorbed on $Co_4$ cluster as well as on the $Co_4/Al_2O_3$ system, the weakness of C–O bond results in its elongation and the decrease of bond-dissociation energy. The adsorption energies of CO on $Co_4/Al_2O_3$ are lower than those on $Co_4$ cluster alone. These results indicate the role of the alumina support: the ***d-2*** configuration CO–$Co_4/Al_2O_3$ is the most favorable configuration due to the lowest adsorption energy, while the dissociation energy of the C–O bond in this configuration is highest. Thus, loading $Co_4$ on the $Al_2O_3$ surface leads to an increase in the CO adsorption ability but hinder the dissociation of C–O bond. In another words, hydrogenation of CO on $Co_4/Al_2O_3$ likely leads to the formation of oxygen-containing compounds. Notably, the adsorption of CO on $Co_4/Al_2O_3$ results in the formation of d-2 configuration. In this configuration the number of unpaired electrons is eight, while in the initial $Co_4/Al_2O_3$ system there are 10 unpaired electrons. Hence, two reduced electrons involve and to form a covalent bond between adsorbent and the CO molecule.

The obtained results regarding to the adsorption ability of CO on $Co_4/Al_2O_3$ show that $Co_4/Al_2O_3$ can catalyze the hydrogenation of CO into alcohols.

### 3.2.2. Adsorption of $H_2$ on $Co_4$ and $Co_4/Al_2O_3$ Systems

It has been established that for adsorption of hydrogen on the transition-metal surfaces the dissociative chemisorption is the preferred [50,51]. Our calculated results also revealthat $H_2$ dissociative adsorptions on $Co_4$ cluster and $Co_4/Al_2O_3$ system are favorable due to the lower adsorption energy (in comparison to that for molecular adsorption (see Supplementary, Table S4). The hydrogen adsorption on $Co_4/Al_2O_3$ releases 280.7 kJ mol$^{-1}$. The H–H distance is elongated from 0.776 Å (in the isolated molecule) to 3.145 Å (in the adsorption structure). Dissociated hydrogen atoms are simultaneously bound to two metal atoms on the surface of catalysts and thus will come to react with adsorbed CO molecule in the next reaction steps.

### 3.2.3. Adsorption of CO and $H_2$ on $Al_2O_3$ System

For comparison purposes, the adsorption of CO and $H_2$ on a pure $Al_2O_3$ support is also estimated. The calculations show that CO molecule is favorably adsorbed horizontally on the $Al_2O_3$ (104) surface at a distance of about 2.6 Å with an adsorption energy of $-59.8$ kJ mol$^{-1}$. The $E_{ads}$ for $H_2$ adsorption on $Al_2O_3$ is $-32.7$ kJ mol$^{-1}$ and the closest distance of $H_2$ to the $Al_2O_3$ surface is 2.3 Å (see Supplementary, Table S5) which reflects a physisorption in nature. These results indicate that $Al_2O_3$ can play a support role only. It interacts with the $Co_4$ sites to enhance the adsorption efficiency of Co toward CO and $H_2$.

### 3.3. Preliminary Investigation on the Process of CO Hydrogenation over $Co_4/Al_2O_3$ Catalyst to Methanol

We consider the first step in the CO conversion process to methanol, this step involves the interaction between CO and $H_2$ adsorbed molecules.

A schematic diagram of the suggested possible reaction pathway for the $CH_3OH$ synthesis via the hydrogenation of CO on the $Co_4/Al_2O_3$ catalyst is presented in Figure 4a,b.

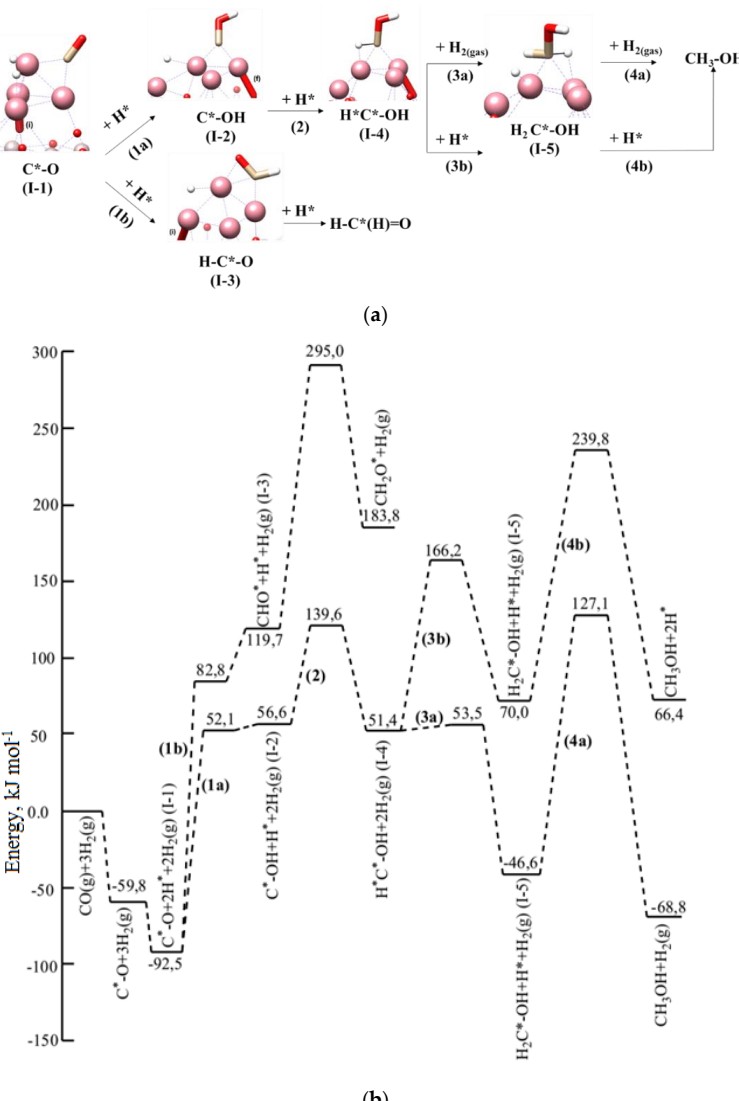

**Figure 4.** The possible reaction pathways for the $CH_3OH$ synthesis via the hydrogenation of CO on the $Co_4/Al_2O_3$ catalyst (colors: bronze—carbon atom, red—oxygen atom, white—hydrogen atoms, violet—cobalt atoms), symbol "*" denotes the adsorbed species. Potential energy diagram obtained from density functional theory (DFT) calculations for $CH_3OH$ synthesis via the hydrogenation of CO on the $Co_4/Al_2O_3$ catalyst, symbol "*" denotes the adsorbed species.

The reaction energy (ΔE) and the activation energy ($E_a$) of each stage are defined as:

$$E_a = E_{TS} - E_{IS}$$

$$\Delta E = E_{FS} - E_{IS}$$

where $E_{TS}$, $E_{IS}$, and $E_{FS}$ are the total energies of the transition states (TS), the initial (IS) and final structures (FS), respectively.

The transition state structures of CO hydrogenation on $Co_4/Al_2O_3$ are presented in Table S6 of Supplementary. The calculated ΔE and $E_a$ for each step are summarized in Table 3.

**Table 3.** The calculated ΔE and $E_a$ for the $CH_3OH$ synthesis reaction pathways.

| Steps | ΔE (kJ mol$^{-1}$) | $E_a$ (kJ mol$^{-1}$) |
|-------|--------|--------|
| 1a | 149.10 | 144.6 |
| 1b | 212.2 | 175.3 |
| 2 | −5.2 | 83.0 |
| 3a | −97.8 | 2.1 |
| 3b | 18.6 | 114.8 |
| 4a | −22.4 | 173.5 |
| 4b | 47.8 | 221.2 |

Starting from the configuration I-1 with adsorbed CO and H species on the $Co_4/Al_2O_3$ surface (Figure 4a,b), the CO hydrogenation may firstly occur through two pathways: the adsorbed CO reacts with the neighboring adsorbed atomic H to generate the C*-OH (I-2) or H-C*-O (I-3) species (the asterisk denotes the adsorbed species). Optimized structures of I-1, I-2 and I-3 are presented in Figure 5. A comparison of the energies of formation (stability) and activation energies of COH and HCO formation by CO hydrogenation shows that the C*-O-H formation is energetically more favorable than H-C*=O formation. This finding is due to the lower activation energy (lower by 30.7 kJ mol$^{-1}$) and lower reaction energy (lower by 63.1 kJ mol$^{-1}$) of COH formation. Thus, CO hydrogenation is likely to proceed via the COH pathway. If one considered the I-3 configuration, the remaining adsorbed hydrogen atom H* may interact with the C atom of H-C*= O species to form formaldehyde –HC(H)=O.

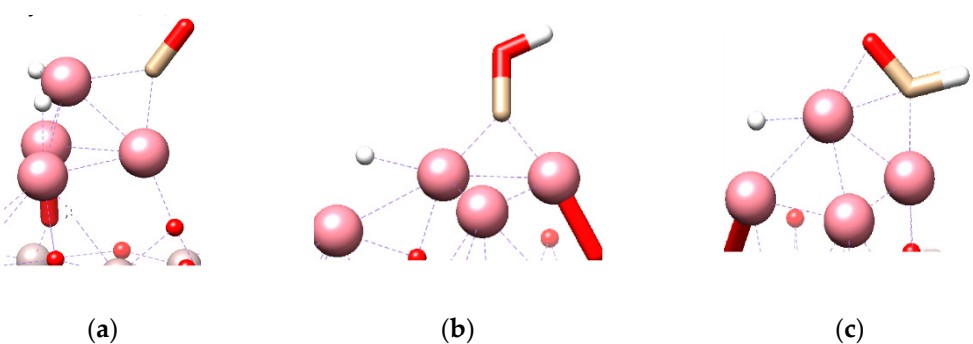

| (a) | (b) | (c) |

**Figure 5.** The optimized structures of I-1 (**a**), I-2 (**b**), and I-3 (**c**) (colors: brown—carbon atom, red—oxygen atom, white—hydrogen atoms, violet—cobalt atoms; symbols (i) and (f) denote the initial and final.

The formation of C*-O-H induced a remarkable change in the geometry of $Co_4$. The $Co_4$ tetrahedron is totally broken and a new $Co_4$ system is formed which is stabilized by the formation of new Co–O bond. The initial Co–O bond in I-1 structure (denoted as **i** in Figure 5) has been broken, a new Co–O (in $Al_2O_3$) bond is formed in I-2 structure (denoted as **f** in Figure 5) and there is no change in the Co–O bond in I-3 configuration. This change in structure of $Co_4$ can be attributed to the $Al_2O_3$ support, which stabilizes (strengthens) the $Co_4$ catalyst during the reaction due to the

formation of Co–O bonds. This stabilization may explain for the difference of our result in comparison to Zuo et al.'s calculation [36]. Zuo et al. study the CO hydrogenation over $Cu_2$ cluster adsorbs onto hydroxylated $Al_2O_3$ (110) surface in liquid paraffin. In their work, methanol was synthesized through the intermediate CHO rather than COH. Our calculation was based on a cluster of $Co_4$ adsorbed onto $Al_2O_3$ (104) surface. However, our finding is in agreement with the work done by Andersen et al. [15], who study CO hydrogenation over the $MoS_2$(100) surfaces using DFT calculations. Their results also indicated that the COH/HCOH route is more favorable than the HCO/$H_2$CO route on the Mo edge surface. The highest activation barrier in the COH/HCOH route is only 134.1 kJ mol$^{-1}$, which is lower than the barrier of 161.1 kJ mol$^{-1}$ for the step forming HCO. These values agree well with our calculations.

From I-2, C*-OH is subsequently hydrogenated to form H-C*-OH (I-4) through interaction of one adsorbed hydrogen atom with C* of C*-OH:

$$H^* + C^*\text{-}OH \rightarrow H\text{-}C^*\text{-}OH$$

Energetically, this process is more dominated because it is exothermic with a negative reaction energy of $\Delta E$ ($-5.2$ kJ mol$^{-1}$). The activation energy is determined to be 83.0 kJ mol$^{-1}$.

Experimentally it is well know that at low hydrogen pressure the methanol formation follows the Langmuir–Hinshelwood mechanism, in which both reactants are adsorbed on the catalytic surface and the interactions between adsorbed species. However, at high hydrogen pressure, the reaction might occur via the Eley–Rideal mechanism, in which one reactant in gas phase will come to interact with other adsorbed reactant [14,52,53]. Based on this observation we predict two possible pathways to form methanol from H-C*-OH (I-4):

(i)   In one route (steps 3a-4a), attack of hydrogen molecule from the gas phase with H-C*-OH (I-4) produces $H_2C^*$-OH (I-5). This route is corresponding to the Eley–Rideal mechanism:

$$H_{2\ (gas)} + H\text{-}C^*\text{-}OH \rightarrow H_2C^*\text{-}OH + H^* \text{ (step 3a)}$$

The optimized structures of I-4 and I-5 are presented in Figure 6.

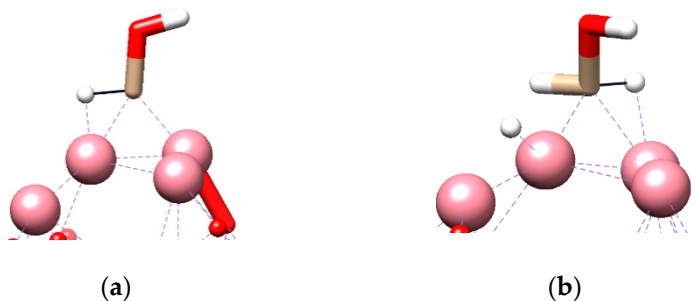

(**a**)                                        (**b**)

**Figure 6.** The optimized structures of I-4 (**a**) and I-5 (**b**).

The formed $H_2C^*$-OH (I-5) then subsequently undergoes hydrogenation by another $H_2$ from the gas phase producing $H_3C^*$-OH and it will desorb to $CH_3OH$ product (step 4a).

$$H_{2\ (gas)} + H_2\ C^*\text{-}OH \rightarrow H_3C^*\text{-}OH + H^* \rightarrow CH_3OH_{gas}$$

(ii)  In the other route (3b-4b), one adsorbed hydrogen atom on the catalyst surface comes to combine with H-C*-OH (I-4) forming $H_2C^*$-OH (I-5). The $H_2C^*$-OH fragment further reacts with another adsorbed hydrogen atom and followed by desorption to release $CH_3OH$ (step 4b). This route

is corresponding to the Langmuir–Hinshelwood mechanism and also has been suggested and calculated by several researchers [15,22,30,32]:

$$H^* + H\text{-}C^*\text{-}OH \rightarrow H_2C^*\text{-}OH \text{ (step 3b)}$$

$$H^* + H_2C^*\text{-}OH \rightarrow H_3C^*\text{-}OH \rightarrow CH_3OH_{gas} \text{ (step 4b)}$$

The formation of methanol by the direct attack of a hydrogen gas molecule to I-4 is energetically favorable because it is exothermic with a reaction energy of $-97.8$ kJ mol$^{-1}$ and an activation energy of 2.1 kJ mol$^{-1}$. Meanwhile, the hydrogenation steps forming $H_2C^*\text{-}OH$ and $H_3C^*\text{-}OH$ through adsorbed hydrogen atoms are kinetically limited due to the higher activation barrier (114.8 kJ mol$^{-1}$).

These calculated results suggest that, at low pressure of $H_2$ and CO (meaning at low CO and $H_2$ surface coverages), the $CH_3OH$ formation will proceeds via consecutive hydrogenation of $C^*$-O fragments by adsorbed hydrogen atom on the catalyst surface to form intermediate species $C^*\text{-}OH$, $H\text{-}C^*\text{-}OH$, $H_2C^*\text{-}OH$ and finally, methanol $CH_3OH$. This would be also consistent with the Langmuir–Hinshelwood mechanism.

However, at high pressure of $H_2$, the reaction pathway will follow the Eley–Rideal mechanism where adsorbed $C^*OH$ and $HC^*OH$ species react with molecules $H_2$ from the gas phase in the slow steps of the reaction (steps 3a-4a). The lower activation energies of these steps (3a-4a) compare to those of steps (3b-4b) and the negative values of reaction energies ($-97.8$ and $-22.4$ kJ mol$^{-1}$ indicating an exothermic nature) calculated by our work correspond well to thermodynamic and kinetic nature of the studied reaction: methanol productivity and rate formation predominates at high pressure.

Generally, In the CO hydrogenation the main reactions which accompany with the methanol synthesis (R1) are the Fischer–Tropsch process (R2) and water–gas shift (WGS) reaction (R3):

$$CO + 2H_2 \rightleftharpoons CH_3OH \tag{R1}$$

$$(2n + 1)\, H_2 + n\, CO \rightarrow C_nH_{2n+2} + n\, H_2O \tag{R2}$$

$$CO + H_2O \rightleftharpoons CO_2 + H_2 \tag{R3}$$

These reactions are not independent, and any one of them can be expressed as a linear combination of the other two. Also conversion to methanol is limited by equilibrium of the reactions taking place. The synthesis of $CH_3OH$ from syngas is a equilibrium limited process. It is a high-pressure and exothermic reaction. Increasing pressure results in the reaction of $CO+2H_2 \rightleftharpoons CH_3OH$ shifting towards the right hand side. On the other hand, the CO conversion, the hydrocarbon formation through Fischer–Tropsch (FT) reaction and the production of $CO_2$ from the WGS process significantly increase with increasing temperature, as a result the $CH_3OH$ selectivity decreases. Therefore, it is necessary to conduct the ($2H_2 + CO$) reaction at high pressures rather than at high temperatures to obtain high $CH_3OH$ yields.

Ding et al. [52] and Wu et al. [14] studied the kinetics of CO hydrogenation into methanol and found that at high temperature ($>330\,°C$) the alcohol yield was not consistent with the conversion of CO because at these temperatures the hydrocarbon production is more selective and the WGS reaction becomes significant. The methanol selectivity monotonically decreased with increasing temperature. But increasing the pressure increases the selectivity of methanol, and higher alcohols.

Ladera et al. [53] investigated the hydrogenation of $CO_2$ into methanol over Ga-doped $Cu/ZnO/ZrO_2$ catalysts Their investigation shows that the $CH_3OH$ yield is increased upon increasing the $H_2$ pressure but levels off with the $CO_2$ pressure. The authors suggested that $CH_3OH$ formation obeys the Eley–Rideal mechanism whereby adsorbed $CO_2$ species ($CO_2^*$) react with $H_2$ molecule in the slow step of the reaction.

## 4. Conclusions

In the present study, the elementary steps of CO hydrogenation leading to the formation of $CH_3OH$ on $Co_4/Al_2O_3$ were analyzed using the density functional theory calculations. A 4-atom Co cluster, $Co_4$, was located on the $Al_2O_3$ (104) surface. The calculated results indicated that the formation of $Co_4/Al_2O_3$ systems is accompanied by the formation of chemical bonds between Co and O atoms. CO is more readily absorbed on $Co_4/Al_2O_3$ than on alone a $Co_4$ cluster. In the CO hydrogenation process, methanol was synthesized through the intermediates C*-OH, H-C*-OH, and $H_2$C*-OH:

$$H_2 \rightarrow H^* + H^*$$

$$H^*\text{-}H^* + C^*\text{-} O \rightarrow H^* + C^*\text{-OH}$$

$$H^* + C^*\text{-OH} \rightarrow H\text{-}C^*\text{-OH}$$

$$H_2 + H\text{-}C^*\text{-OH} \rightarrow H_2C^*\text{-OH} + H^*$$

$$H_2 + H_2C^*\text{-OH} \rightarrow H_3C\text{-OH} + H^*$$

Our calculations show that the formation of C*-O-H induced a significant change in the geometry of $Co_4$ and our calculated results show that the interaction between $Al_2O_3$ support and Co catalyst plays a key role in the catalytic hydrogenation of CO to methanol.

The studies have also suggested that the initial steps are kinetically difficult although the COH/HCOH pathway is slightly favored more than $HCO/H_2CO$ pathway. $CH_3OH$ synthesis on $Co_4/Al_2O_3$ proceeds through CO reaction with hydrogen via either an Eley–Rideal or Langmuir–Hinshelwood pathway to form C*O, C*OH, HC*OH, $H_2$C*OH and finally, methanol ($CH_3OH$), with both hydrogenation steps forming C*-OH and final product as rate-limiting.

Our DFT results shed light on the effects of $Al_2O_3$ support and the $H_2$ pressure on the reaction pathways for methanol formation from CO hydrogenation over a Co catalyst. The strong metal–support interaction will lead to high dispersion of active sites. This theoretical study allows us to describe CO hydrogenation reaction in detail and is capable of explaining the key experimental trend of this important but complex reaction.

**Supplementary Materials:** The following are available online at http://www.mdpi.com/2073-4344/9/1/6/s1, **Table S1**. The calculated energy (Ecluster) of Co4 clusters with different number of unpaired electrons. **Table S2**. The calculated interation energy (Eint) between Co4 cluster and $Al_2O_3$. **Table S3.** The calculated adsorption energies ($E_{ads}$) for the adsorption of CO on $Co_4$ and $Co_4/Al_2O_3$ systems. **Table S4.** The calculated adsorption energies ($E_{ads}$) for the adsorption of $H_2$ on $Co_4$ and $Co_4/Al_2O_3$ system. **Table S5.** The calculated adsorption energies ($E_{ads}$) for the adsorption of CO and $H_2$ on $Al_2O_3$ system. **Table S6.** Transition state structures of CO hydrogenation on Co4/$Al_2O_3$.

**Author Contributions:** Conceptualization, L.M.C. and N.N.H.; Methodology, N.N.H. and N.T.T.H.; Formal Analysis, L.M.C. and N.N.H.; Investigation, N.N.H., N.T.T.H. and N.B.L.; Data Curation, N.T.T.H. and N.B.L.; Writing-Original Draft Preparation, N.N.H. and N.T.T.H.; Writing-Review andEditing, L.M.C. and N.T.T.H.; Visualization, L.M.C. and N.T.T.H.; Project Administration, L.M.C.

**Funding:** This research was funded by the Vietnam Ministry of Education and Training, grant number B2015-17-69.

**Conflicts of Interest:** The authors declare no conflict of interest. The founding sponsors had no role in the design of the study; in the collection, analyses, or interpretation of data; in the writing of the manuscript; and in the decision to publish the results.

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
