# Peer review of "Conversion of Carbon Monoxide into Methanol on Alumina-Supported Cobalt Catalyst: Role of the Support and Reaction Mechanism—A Theoretical Study"

_catalysts, doi:10.3390/catal9010006_

Reviewer 1 Report

This manuscript has been substantially improved since the previous version. The methods used are now described in considerably more detail. Much has been added to both the discussion and the conclusions (which are now quite good, in fact), and most, but not all of my previous suggestions have been addressed. Some substantial points still must be addressed, however, before it can be published:

Although some progress has been made on the English language used, there are still too many errors on every page to enumerate. The new (highlighted) parts are particularly poorly written. If the authors have not used a professional language editing service, I am certain that the editors will help them identify an appropriate one. If they have used a service in the past, they should find a different one, as this one does not appear to have caught all errors.   

The authors suggest that they have replaced the old figure 4 with a better one. This is not better. Nearly every computational paper that reports on a reaction mechanism uses a standard figure type. I suggest the authors see, for example, Asgharzade, et al, Computational and Theoretical Chemistry 1104, PP 47-55 (2017). This paper describes shows the energies of every component in the mechanism in a graphical fashion, with optimized structures included, allowing the reader to quickly digest the results. This work is standard, and should be followed.

In ll. 170-178, tetrahedral geometry is eliminated because it has a weaker energy of bonding, but this should not eliminate it from the discussion. It could still occur, and could have better catalytic properties. 

In addition, there are minor issues still to be addressed:

The abstract should not begin with "In present work". We know that abstracts describe the present work. 

Particularly on page 2, but likely elsewhere as well, there are many chemical compound formulas whose subscripts are missing.

The mechanism described in l. 45 would be better presented graphically as a scheme. 

A "dissociate adsorption" error remains in line 234.

In line 237, the negative sign is unnecessary and confusing when used within the construct "releases -2807 kJ/mol of energy".The units are also oddly formatted.

In Figure 5, the (i) and (f) should also be explained in the caption, not just in the text. 

Throughout the discussion, the notation C*=O is used. What is the meaning of the asterisk? 

In line 292, and then again in lines 303-304, statements are made about what happens next. On what is this assertion based? Is this a reference to experimental results, or have the authors in some way calculated this. If the former, a citation is appropriate. If the latter, a clearer explanation is needed. 

Correction of these remaining points will result in a publication quality manuscript, in my view.

Author Response

We would like to thank the reviewer for his thoughtful and thorough review and believe his input has been invaluable to strengthen our manuscript.

1. This manuscript has been substantially improved since the previous version. The methods used are now described in considerably more detail. Much has been added to both the discussion and the conclusions (which are now quite good, in fact), and most, but not all of my previous suggestions have been addressed. Some substantial points still must be addressed, however, before it can be published:

Although some progress has been made on the English language used, there are still too many errors on every page to enumerate. The new (highlighted) parts are particularly poorly written. If the authors have not used a professional language editing service, I am certain that the editors will help them identify an appropriate one. If they have used a service in the past, they should find a different one, as this one does not appear to have caught all errors.  

Response: we have again try to correct English in the whole manuscript

2. The authors suggest that they have replaced the old figure 4 with a better one. This is not better. Nearly every computational paper that reports on a reaction mechanism uses a standard figure type. I suggest the authors see, for example, Asgharzade, et al, Computational and Theoretical Chemistry 1104, PP 47-55 (2017). This paper describes shows the energies of every component in the mechanism in a graphical fashion, with optimized structures included, allowing the reader to quickly digest the results. This work is standard, and should be followed.

Response : We have added Fig 4b (Potential energy diagram obtained from DFT calculations for CH3OH synthesis via the hydrogenation of CO on the Co4/Al2O3 catalyst, symbol “*” denotes the adsorbed species) to address this comment.

Figure 4b. Potential energy diagram obtained from DFT calculations for CH3OH synthesis via the hydrogenation of CO on the Co4/Al2O3 catalyst, symbol “*” denotes the adsorbed species

3. In ll. 170-178, tetrahedral geometry is eliminated because it has a weaker energy of bonding, but this should not eliminate it from the discussion. It could still occur, and could have better catalytic properties.

Response: No, the rhombus geometry (not the tetrahedral) is eliminated from discussion because it is less stable than the tetrahedral one. Our aim in this study is only to work out the role of the support and determine the possible reaction mechanism. So we do not perform/compare the catalytic properties of different clusters (tetrahedral or rhombus Co4, or Co4 and other Con (n=2,3 …)). It will be studied in our following works.

In addition, there are minor issues still to be addressed:

4. The abstract should not begin with "In present work". We know that abstracts describe the present work.

Response: The correction has been made

5. Particularly on page 2, but likely elsewhere as well, there are many chemical compound formulas whose subscripts are missing.

Response: The corrections are made

6. The mechanism described in l. 45 would be better presented graphically as a scheme.

Response: we have re-written as follow:

CO + H → HCO

HCO + H → H2CO

H2CO + H → H2COH

H2COH → OH + CH2

CH+ H → CH3

CH3  + H → CH4

7. A "dissociate adsorption" error remains in line 234.

Response: We have  re-written “dissociative adsorption”

8. In line 237, the negative sign is unnecessary and confusing when used within the construct "releases -280.7 kJ/mol of energy".The units are also oddly formatted.

Response:  The correction has been made. The units are also reformated (kJ.mol-1  => kJ mol-1)

9. In Figure 5, the (i) and (f) should also be explained in the caption, not just in the text.

Response: We have added information “Figure 5.The optimized structures of I-1 (a), I-2 (b), and I-3 (c) (colors: brown – carbon atom, red – oxygen atom, white – hydrogen atoms, violet – cobalt atoms, symbols (i) and (f) denote the initial and final ”

10. Throughout the discussion, the notation C*=O is used. What is the meaning of the asterisk?

Response: Symbol “*” denotes the adsorbed species (as mentioned above on the Fig 4a and 4b). We have added the following sentence into the text “to generate the C*-OH (I-2) or H-C*-O (I-3) species (the asterisk denotes the adsorbed species)” to address this comment

11. In line 292, and then again in lines 303-304, statements are made about what happens next. On what is this assertion based? Is this a reference to experimental results, or have the authors in some way calculated this. If the former, a citation is appropriate. If the latter, a clearer explanation is needed. 

Response: It is a reference to experimental results. Normally when doing calculation all interaction possibilities should be considered. However, we may shorter the calculation base on the experimental observations. We have add into the text some discussion

Experimentally it is well know that at low hydrogen pressure the methanol formation follows Langmuir-Hinshelwood mechanism, in which both reactants are adsorbed on the catalytic surface and the interactions between adsorbed species. However, at high hydrogen pressure, the reaction might occur via Eley-Redeal mechanism, in which one reactant in gas phase will come to interact with other adsorbed reactant [52-54]. Based on this observation we predict two possible pathways to form methanol from H-C*-OH (I-4):

In one route (steps 3a-4a), attack of hydrogen molecule from the gas phase with H-C*-OH (I-4) produces H2C*-OH (I-5). This route is corresponding to Eley-Redeal mechanism: H2 (gas) + H-C*-OH ® H2C*-OH + H* (step 3a)

         In other route (3b-4b), one adsorbed hydrogen atom on the catalyst surface comes to combine with H-C*-OH (I-4) forming H2C*-OH (I-5). H2C*-OH fragment further reacts with another adsorbed hydrogen atom and followed by desorption to release CH3OH (stage 4b). This route is corresponding to Langmuir-Hinshelwood mechanism and also this route has been suggested and calculated by several researchers [15, 22, 30, 32]:

H* + H-C*-OH ® H2C*-OH                           (step 3b)

H* + H2C*-OH ® H3C*-OH ® CH3OHgas      ( step 4b)

Reviewer 2 Report

The revised version gives better description on the methods used and on the Introduction section. There are, however, a few points that are not appropriately addressed in this version.

The calculation is based on the SIESTA method/program which is not generally available or commonly used. The authors are suggested to describe its special functionality and advantages. A few successful application of the methods in related studies should be discussed in the Introduction. 

The spin-state problems were addressed by the authors in a seperate document. First of all, this should be included in the text. Secondly, the authors simplify the matter and rely  sole on the SIESTA to handle the difficulty automatically without giving the proof and more detail information. The justfication of the theoretical methods still seems weak.

It is still puzzling that the authors aim to study the "mechanism" but are reluctant to provide the TS information. A couple examples can be discussed  in the main text and others in Supplementary material.

Author Response

We would like to thank the reviewer for very careful and thorough reading of this manuscript and forthe useful comments and constructive suggestions, which help to improve the quality of this manuscript.

The revised version gives better description on the methods used and on the Introduction section. There are, however, a few points that are not appropriately addressed in this version.

1. The calculation is based on the SIESTA method/program which is not generally available or commonly used. The authors are suggested to describe its special functionality and advantages. A few successful application of the methods in related studies should be discussed in the Introduction. 

Response: We have added the following sentence into the text to address this comment “SIESTA has been successfully performed to study on the Fischer-Tropch reaction [39-41] due to its advantages in robust and accurate aspect.”

2. The spin-state problems were addressed by the authors in a seperate document. First of all, this should be included in the text. Secondly, the authors simplify the matter and rely sole on the SIESTA to handle the difficulty automatically without giving the proof and more detail information. The justfication of the theoretical methods still seems weak.

Response: We optimized Co4 in 2 different ways: with fixed spin (10 and 14 unpaired electrons) and relaxed spin (automatically determined by SIESTA, 12 unpaired electrons). The energies of the optimal structures are -2871.13 (10 unpaired electrons), -2872.61 eV (14 unpaired electrons) and -2873.93 eV (relaxed spin - 12 unpaired electrons), respectively. Please see the Supplementary 2. We have also added the following sentences into the text “In our calculations, we have used collinear spin polarized option. The total spin polarization for the optimal structure were automatically determined corresponding to the structure with the lowest energy. Furthermore, the Co4 structure were optimized with the fixed spin corresponding to the presence of 10 or 14 unpaired electrons. The results have confirmed that the Co4 structure with 12 unpaired electrons is the most stable due to the lowest eneergy (see Supplementary-1). ”

3. It is still puzzling that the authors aim to study the "mechanism" but are reluctant to provide the TS information. A couple examples can be discussed  in the main text and others in Supplementary material.

Response: Regarding to TSs, in this report, we want to focus on Ea only. That is the reason why we do not discuss about structures of TSs. However we have supplemented more detail about the structures of the TSs (please see the Supplementary-2 Table 5)and added the following sentence into the text The transition state structures of CO hydrogenation on Co4/Al2O3 are presented in Table 5 of  the Supplementary-2

Reviewer 3 Report

The MS has been mostly re-written and provides more details on DFT simulation, results and discussion. However, the computational details show that the design of DFT calculation was not correct at the beginning (surfaces, vacuum layer ?, unit cell ?) . So I am sorry but I can not accept this MS. 

Author Response

The DFT calculations can be performed for periodic as well as for non periodic systems. In this work we used non periodic system with a large number of atoms. In the manuscript, we have already showed:

- Surface: Al2O3(104)-(2x2x2), that means Al2O3 (104) plane and we used 8 Al2O3 unit cells (2x2x2 in three dimensions).

- Vacuum layer: In this study, no periodic DFT was used. That is the reason why the system is placed in the box of 25 x 25 x 25 angstroms. Placing in a box plays the same role as using vacuum layer.

Reviewer 4 Report

In this work the authors studied through DFT calculations the CO hydrogenation toward methanol on a solid catalyst based on Cobalt dispersed on alumina (Al2O3). A Co4 cluster supported on a slab of oxide was used as model to represent the catalyst. The preferably route for the CO hydrogenation to methanol on this catalyst model was determined, concluding that the hydrogenation occurs through the COH intermediary. The COH formation and the last hydrogenation step are those determining the reaction rate. Support effects were also considered in the study.

In my opinion this work has enough novelty to be published in Catalysts but the authors have to clarify some aspects before it deserves publication.

My main objection is how reliable is the catalyst model used in this work?. The authors used a model constituted by four cobalt atoms on a oxide slab, but are the real catalysts for the CO hydrogenation on Co/Al2O3 based on a small number of metallic atoms dispersed on a support?. This has to be clarified and the effect in the results of the small size of the Co cluster has to be quantified.

.The format of the References has to checked again (even some surnames are not corrected written).

Author Response

We would like to thank the reviewer for very careful and thorough reading of this manuscript and for the useful comments and constructive suggestions, which help to improve the quality of this manuscript.

In this work the authors studied through DFT calculations the CO hydrogenation toward methanol on a solid catalyst based on Cobalt dispersed on alumina (Al2O3). A Co4 cluster supported on a slab of oxide was used as model to represent the catalyst. The preferably route for the CO hydrogenation to methanol on this catalyst model was determined, concluding that the hydrogenation occurs through the COH intermediary. The COH formation and the last hydrogenation step are those determining the reaction rate. Support effects were also considered in the study. In my opinion this work has enough novelty to be published in Catalysts but the authors have to clarify some aspects before it deserves publication.

- My main objection is how reliable is the catalyst model used in this work?. The authors used a model constituted by four cobalt atoms on a oxide slab, but are the real catalysts for the CO hydrogenation on Co/Al2O3 based on a small number of metallic atoms dispersed on a support?. This has to be clarified and the effect in the results of the small size of the Co cluster has to be quantified.

Response: Yes, we want to simulate the catalytic system where metallic atoms are well dispersed that mean they exist as small size clusters. According to Datta et al. (Datta, S., Kabir, M., Ganguly, S., Sanyal, B., Saha-Dasgupta, T., & Mookerjee, A. (2007). Structure, bonding, and magnetism of cobalt clusters from first-principles calculations. Physical Review B, 76(1). doi:10.1103/physrevb.76.014429) among small clusters Con (n=2-10) Co6 and Co9 are the most stable. Co4 is more stable than Co2, Co3, Co5, Co7, Co9 and Co10. However due to the limitation of calculation (very demanding for big system), the Co4 was chosen.

- The format of the References has to checked again (even some surnames are not corrected written).

Response: We have checked

Round  2

Reviewer 2 Report

I agree with the authors that the manuscript is publishable in present form.

Still, I don't quite get it why it is so difficult to discuss TS structure in the manuscript. Ea can be misleading if the TS structures are unresonable. But I don't insist.

Author Response

We would like to thank the reviewer for his useful comments. We have checked English language and style of our manuscript as suggested.

Reviewer 3 Report

I have carefully read through the new version and the reply letter. I would like to keep my decision on the basis of err in DFT set up. Going deep into the details, the MS is lacking of model consistency. I do not agree that they can use such a small cluster for the model of Al2O3 surface. Al2O3 cluster (without periodic as mentioned in this work) is another system.

Author Response

We appreciate the feedback from the reviewer. 

We would like to keep our opinion on using the non-periodic model of Al2O3 consisting of a large enough number of atoms (8 (2*2*2) unit cells). It is suitable to figure out the role of the alumina support on the conversion of only 1 CO molecule on a small adsorbed cobalt cluster. 

Reviewer 4 Report

The authors have satisfactory answered my principal objection but some minor points are not correct yet.

The authors have to check the references again and change the format for the Catalyst style. .Surnames as Ramíreza or Sáncheza are not correctly written yet.

Mechanism Eley-Redeal should be written Eley-Rideal.

Author Response

We would like to thank the reviewer for very careful and thorough reading of this manuscript.

Our response follows (the reviewer’s comments are in italics).

The authors have satisfactory answered my principal objection but some minor points are not correct yet.

The authors have to check the references again and change the format for the Catalyst style:

We have checked the References’style. In some cases, we can not determine the first name, surname of the authors. So we write the original names.

Surnames as Ramíreza or Sáncheza are not correctly written yet: Thank you. The correction has been made.

Mechanism Eley-Redeal should be written Eley-Rideal: The correction has been made

This manuscript is a resubmission of an earlier submission. The following is a list of the peer review reports and author responses from that submission.

Round  1

Reviewer 1 Report

This manuscript describes a comprehensive evaluation of the conversion of the hydrogenation of carbon monoxide over a cobalt catalyst supported on an alumina surface. The manuscript begins with a thorough review of the work available on other catalysts, and places the present research in strong context. Overall, the methods described appear sound, but more detail is needed in places (even if it is relegated to supplemental information). The results are clearly explained, but in some cases are difficult to evaluate because insufficient description of methods is provided. I believe that this work is fundamentally very solid, and can be excellent with some minor revisions. Although the language errors do not drastically interfere with understanding, the writing should be reviewed extensively by a native speaker before publication.

Some specific comments:

1. lines 112 and 115 make reference to (chi)-Al2O3, which I believe is likely intended to be (gamma)-Al2O3.

2. The computational methods section requires more citation. Every method used should be cited. For example (but not only!), PBE/DZP should have at least one citation (and should probably be described as a functional, not a method) for the functional PBE, and a separate one for DZP, and more details about the pseudo-potentials should be provided, as what is presented is insufficient for another researcher to reproduce the work. What computational package(s) was used to perform the calculations? These are standard things to include in any computational study, and I was quite surprised that the authors did not follow normal practices in describing their theoretical methods.

3. In many cases, such as in line 170, the most stable structure is presented, but it is not clear what the other structures are, or how it was determined to be the most stable. I assume that an extensive search has been performed to identify the most stable structure, and that many different local minima were identified. This needs to be available in each case, at least in supplemental information. A table, including energies used to make the determination of “most stable”, should be included. Similar issues occur in line 196 (indicated by what calculations? Where are these results??),

4. Table 2 lists only one set of parameters and thus should not be a table. Further, it is not useful to have these numbers without the context of other numbers against which they can be compared. The reader is not able to evaluate their relevance nor the extent to which they are optimal.

5. In Line 178, total atomic charge is used. There are many ways to evaluate this. Do the authors mean Mulliken charges? If so, they should be aware of the limitations of this method, and should acknowledge those limitations in their discussion.

6. I am not familiar with the use of mononuclear and binuclear as they are used in 200 and 201, and find the terms confusing, given that the mononuclear represents a situation in which two nuclei are bound, and binuclear, three nuclei. If this is standard phrasing, that is fine – I may just not be aware of it.

7. Figure 3 caption has some bold/italic information missing for d-2.

8. Figure 4 needs substantial improvement, but could result in a tremendously powerful figure. When describing a reaction mechanism, where activation energies and delta G values have been calculated, it is conventional to represent this information graphically and quantitatively. This appears instead to simply be a flowchart, but the paper contains everything it needs to produce a figure that shows the relative energy of all reactants, products, and intermediates, and shows the activation energies between them quantitatively, with energy indicated by position along the vertical axis. This will substantially clarify the entire results section in a very powerful way.

9. Line 272, the notation (2.4) I take to mean the energy difference between 2 and 4, but that might be better indicated with an arrow.

10. Page 8, the bullets in addition to an outline are redundant.

11. Line 283: this is the second place where “dissociate adsorbed” is used, but it is not clear what the phrase means, and it is certainly grammatically incorrect in any situation. Consider revising and clarifying.

12. The conclusions could be more forward-thinking and large-picture oriented. Currently, a well-written summary is provided, but little is done to reconnect the work to the context presented in the introduction or to offer insights into future uses for this work. A little additional text here is warranted.

I believe that these are minor and not major revisions, but they are important and will take this from a marginal paper to what I would consider a truly excellent one.

Author Response

1.        lines 112 and 115 make reference to (chi)-Al2O3, which I believe is likely intended to be (gamma)-Al2O3:

Response: we have corrected and rewritten a part of the Introduction

2.        The computational methods section requires more citation. Every method used should be cited. For example (but not only!), PBE/DZP should have at least one citation (and should probably be described as a functional, not a method) for the functional PBE, and a separate one for DZP, and more details about the pseudo-potentials should be provided, as what is presented is insufficient for another researcher to reproduce the work. What computational package(s) was used to perform the calculations? These are standard things to include in any computational study, and I was quite surprised that the authors did not follow normal practices in describing their theoretical methods.

Response: we have supplemented more detail about computational methods. It is highlighted in the new manuscript. In the revised version, we have described all specific information related to our calculations.

3.        In many cases, such as in line 170, the most stable structure is presented, but it is not clear what the other structures are, or how it was determined to be the most stable. I assume that an extensive search has been performed to identify the most stable structure, and that many different local minima were identified. This needs to be available in each case, at least in supplemental information. A table, including energies used to make the determination of “most stable”, should be included. Similar issues occur in line 196 (indicated by what calculations? Where are these results??)

Response:We have added some data in the manuscript and attached Supplementary to consolidate our computational results. In the Supplementary all the calculated optimized structures along with the energy characterizations are presented.

4.        Table 2 lists only one set of parameters and thus should not be a table. Further, it is not useful to have these numbers without the context of other numbers against which they can be compared. The reader is not able to evaluate their relevance nor the extent to which they are optimal.

Response: we have rewritten this paragraph

5.        In Line 178, total atomic charge is used. There are many ways to evaluate this. Do the authors mean Mulliken charges? If so, they should be aware of the limitations of this method, and should acknowledge those limitations in their discussion:

Response: The atomic charges used in this work are Voronoi charges. In the rewritten paragraph Model and Computational Methods, we have added the following sentences “Besides, the atomic partial charges, estimated by means of the Voronoi deformation density (VDD) method were reported. The VDD method avoids the problems inherent to basis set based schemes and provides meaningful charges that conform to chemical experience [42]”.

6.        I am not familiar with the use of mononuclear and binuclear as they are used in 200 and 201, and find the terms confusing, given that the mononuclear represents a situation in which two nuclei are bound, and binuclear, three nuclei. If this is standard phrasing, that is fine – I may just not be aware of it.

Response: we have replaced the terms “mononuclear and binuclear” to address this comment

7.        Figure 3 caption has some bold/italic information missing for d-2:

Response: The correction has been made.

8.        Figure 4 needs substantial improvement, but could result in a tremendously powerful figure. When describing a reaction mechanism, where activation energies and delta G values have been calculated, it is conventional to represent this information graphically and quantitatively. This appears instead to simply be a flowchart, but the paper contains everything it needs to produce a figure that shows the relative energy of all reactants, products, and intermediates, and shows the activation energies between them quantitatively, with energy indicated by position along the vertical axis. This will substantially clarify the entire results section in a very powerful way.

Response: We have replaced Fig 4 by the better one. In this paper we focus on the effect of the alumina support to the activity of Co sites. The figure is only the illustration we used to show the role of the support and consider only the Eact to evaluate the effect (if any) of the support on the reaction pathways. The analysis of nature of the TS and detail of reaction mechanism will appear in the coming our paper.

9.        Line 272, the notation (2.4) I take to mean the energy difference between 2 and 4, but that might be better indicated with an arrow:

Response: we have corrected

10.        Page 8, the bullets in addition to an outline are redundant:

Response: The correction has been made

11.        Line 283: this is the second place where “dissociate adsorbed” is used, but it is not clear what the phrase means, and it is certainly grammatically incorrect in any situation. Consider revising and clarifying

Response: we have rewritten this sentence and replaced “dissociate adsorbed” by “dissociative adsorption/chemisorption” which means molecule is dissociated when adsorbed on the catalyst surface.

12.        The conclusions could be more forward-thinking and large-picture oriented. Currently, a well-written summary is provided, but little is done to reconnect the work to the context presented in the introduction or to offer insights into future uses for this work. A little additional text here is warranted.

Response: We have rewritten the Conclusions

Reviewer 2 Report

The manuscript by Ha et al. describes a computational study on the hydrogenation of CO to methanol. While the topic is hot and important, the study, in my opinion, does not contribute significantly to this field. Specific comments are:

The Introduction gives a long list of recent computational studies. However, little relationship to the experiments is mentioned. In depth review of the appropriate methods and the validity and importance of previous studies is lacking.

The most serious problem is on the theoretical treatment. Why the specific method PBE/DZP was used? What programs? The description of  pseudopotential is not appropriate.

The treatment of Co4 cluster is problematic. It is chemically very unreasonable that  "Co4 has twelve unpaired electrons". The various spin states must be calculated and analyzed. The possible oxidation states might also need to be taken into consideration. Thus, all the results related to the Co4 cluster are questionable.   

The characterization and the nature of the TS is not discussed.

Author Response

1.  The Introduction gives a long list of recent computational studies. However, little relationship to the experiments is mentioned. In depth review of the appropriate methods and the validity and importance of previous studies is lacking.

Response: we have significantly reconstructed and rewritten the Introduction with concerning to your suggestions.

2.  The most serious problem is on the theoretical treatment. Why the specific method PBE/DZP was used? What programs? The description of pseudopotential is not appropriate.
Response: We have rewritten the section Models and Computational methods to address this comment. In this manuscript it is highlighted. In the revised version, we have described all specific information related to our calculations.

3.  The treatment of Co4 cluster is problematic. It is chemically very unreasonable that "Co4 has twelve unpaired electrons". The various spin states must be calculated and analyzed. The possible oxidation states might also need to be taken into  consideration. Thus, all the results related to the Co4 cluster are questionable.

Response: Our calculations use SIESTA code. SIESTA can read a system with different spin structure by adapting the information to the currently selected spin multiplicity, averaging or splitting the spin components equally, as needed. This may be used to greatly increase convergence. In our calculations, we have used Spin polarized option performing a calculation with collinear spin (two spin components). And the total spin polarization for the optimal structure (Qup-Qdown) were automatically determine corresponding to the structure with the lowest energy. For the Co4 (Qup-Qdown)=12 i.e. Co4 has twelve unpaired electrons with spin up.

4.  The characterization and the nature of the TS is not discussed.

Response: Our work focuses on the effect of the alumina support on the conversion of CO. The proposed mechanism is only the illustration we used to show the role of the support. Of course, there are a lot of reaction pathways to convert CO into methanol. However the purpose of our study is not to study the full potential energy surface of the reaction to examine which pathway is more favorable. It is very complicated problem due to the size of the studied system. With regards to the length of the article, we do not focus on analysis the nature of the TS. And just only consider the Eact to evaluate the effect (if any) of the support on the mechanism of reaction.

Reviewer 3 Report

The following minor changes are required:

In the abstract the authors should be more specific then "density function theory" was used.

The first paragraph of the introduction should be deleted (appears to be instructions).

The authors keep writing "density functional theory (DFT)".  defining DFT once is sufficient.

Other typographical and grammatical errors are noted on the attached annotated copy of the manuscript.

Author Response

1.      In the abstract the authors should be more specific then "density function theory" was used. The first paragraph of the introduction should be deleted (appears to be instructions).:

Response: The correction has been made.

2.        The authors keep writing "density functional theory (DFT)". defining DFT once is sufficient

Response: We have rewritten and reconstructed the Introduction. An in Introduction we have added the following sentences to address this comment “Over the past few decades, theory starts to be recognized as an essential tool in optimization of catalysts for energy applications [29,30]. Among theoretical methods used to study the electronic structure and properties of many-body systems, in particular catalytic reactions, density functional theory (DFT) is recognized as a versatile and effective method. The fundamental aspect of the DFT method is that the properties of many- electron systems can be determined using the electron density which depends only on the three Cartesian variables. Using the DFT approach, the computational accuracy can be significantly increased without the additional increase in the computational costs. Advances in DFT make it possible to describe catalytic reactions at surfaces with the detail and accuracy required for computational results to compare with experiment in a meaningful way [31] Such understanding allows the theoretical optimization for better catalysts”

3.        Other typographical and grammatical errors are noted on the attached annotated copy of the manuscript

Response: The correction has been made.

Reviewer 4 Report

The topic of CO conversion to methanol is very interested in Catalyst research. However, there are many weak points in the manuscript (MS):

- The English language needs editing service. 

- No clear on method (software, basis set, PBC, etc..?) and model (PBC or not, size effect ?)

- It seems that the MS needs to be improved extensively in the results and discussion, not just present the data.

Author Response

1.        The English language needs editing service.

Response: The corrections have been made.

2.        No clear on method (software, basis set, PBC, etc..?) and model (PBC or not, size effect ?):

Response: We have rewritten the section Models and Computational methods to address this comment. In this manuscript it is highlighted. In the revised version, we have described all specific information related to our calculations.

3.        It seems that the MS needs to be improved extensively in the results and discussion, not just present the data

Response: We have carefully rewritten and reconstructed our manuscript, extended the results and discussion. A detail discussion has been added. Please find in the revised version of the manuscript.